# Effect of the Helping Babies Breathe Program on Newborn Outcomes: Systematic Review and Meta-Analysis

**DOI:** 10.3390/medicina58111567

**Published:** 2022-10-31

**Authors:** Sergio Agudelo-Pérez, Annie Cifuentes-Serrano, Paula Ávila-Celis, Henry Oliveros

**Affiliations:** School of Medicine, Universidad de La Sabana, Campus Puente del Común, Km. 7, Autopista Norte de Bogotá, Chía 250001, Colombia

**Keywords:** Helping Babies Breath Program, Basic Newborn Resuscitation, infant mortality, asphyxia neonatorum, critical care outcomes

## Abstract

*Background and objectives:* In low- and middle-income countries, the leading cause of neonatal mortality is perinatal asphyxia. Training in neonatal resuscitation has been shown to decrease this cause of mortality. The program “Helping Babies Breathe” (HBB) is a program to teach basic neonatal resuscitation focused on countries and areas with limited economic resources. The aim of the study was to determine the effect of the implementation of the HBB program on newborn outcomes: mortality and morbidity. *Material and Methods:* A systematic review was carried out on observational studies and clinical trials that reported the effect of the implementation in low- and middle-income countries of the HBB program on neonatal mortality and morbidity. We carried out a meta-analysis of the extracted data. Random-effect models were used to evaluate heterogeneity, using the Cochrane Q and I2 tests, and stratified analyses were performed by age and type of outcome to determine the sources of heterogeneity. *Results:* Eleven studies were identified. The implementation of the program includes educational strategies focused on the training of doctors, nurses, midwives, and students of health professions. The poled results showed a decrease in overall mortality (OR 0.67; 95% CI 0.57, 0.80), intrapartum stillbirth mortality (OR 0.62; 95% CI 0.51, 0.75), and first-day mortality (OR 0.70; 95% IC 0.64, 0.77). High heterogeneity was found, which was partly explained by differences in the gestational age of the participants. *Conclusions:* The implementation of the program HBB in low- and medium-income countries has a significant impact on reducing early neonatal mortality.

## 1. Introduction

The worldwide neonatal mortality rate is approximately 19 deaths per 1000 live births [1], of which 90% occur in low-income countries [2]. In this context, the Sustainable Development Goals (SDG) of the World Health Organization (WHO) proposed to end preventable deaths in newborns, reducing neonatal mortality to 12 per 1000 live births by 2030 [1]. During labor and birth, the highest mortality rate is concentrated at 73% of deaths in this period [3,4], mainly due to perinatal asphyxia [5]. 

On the other hand, the training of health personnel in neonatal resuscitation has been shown to be a strategy for reducing mortality and perinatal asphyxia [6,7]. Therefore, the WHO recommended the presence of a person skilled in neonatal resuscitation at all births [8]. So, implementing standardized programs in neonatal resuscitation training for personnel in charge of newborns during birth could reduce neonatal mortality [9].

Nevertheless, the current recommendations of the International Liaison Committee on Resuscitation (ILCOR) are aimed at high-income countries, which makes them difficult to implement in countries with the highest neonatal mortality rate [10]. Therefore, it is necessary to adapt the neonatal resuscitation recommendations to these countries [10]. The American Academy of Pediatrics (AAP), with the support of other agencies, has developed a modified neonatal resuscitation program called Helping Babies Breathe (HBB). It is an evidence-based educational program for low and middle-income countries and areas with limited economic resources, focused on the first minute of life or “golden minute”, to teach respiratory support (Basic Neonatal Resuscitation) with a mask bag, thermoregulation, stimulation, evaluation, and early initiation of breastfeeding [11,12,13].

In some studies, it has been observed that the implementation of the program could influence the reduction of neonatal mortality [14,15], and the economic evaluation of the implementation of the program has been shown to be cost-effective in the prevention of neonatal mortality [16,17]. In this frame, it is important to recognize the importance of training health personnel and the implementation of basic neonatal resuscitation programs as a measure to deal with this problem. The objective of this study was to determine the effect of the implementation of the HBB program on newborn mortality and morbidity.

## 2. Materials and Methods

### 2.1. Literature Search Strategy

A systematic review using the PRISMA-2020 guideline for the identification, screening, and inclusion of studies was conducted. The protocol was registered in PROSPERO (International Prospective Register of Systematic Reviews) with the code CRD 42021264846.

The search was carried out from 1 to 30 June 2021 in the electronic databases Pubmed, EMBASE, LILACS (Literatura Latinoamericana y del Caribe en Ciencia de la Salud), Web of Science, and Cochrane Central Register of Controlled Trials. A manual search was also carried out on Google Scholar, the official website of the HBB program, the personal files of the researchers, and using the snowball method. The search was not restricted by language or year. When needed, additional information and clarified information from data published by individual trial authors were requested.

The search terms used included synonyms or thesauri from the MeSH (Medical Subjects Heading) web dictionaries: newborn, neonate, infant, Helping Babies Breathe, golden minute, neonatal resuscitation, and mortality. The following search strategy was used for Pubmed and was adapted for other electronic databases: ((helping [All Fields] AND (“infant” [MeSH Terms] OR “infant” [All Fields] OR “babies” [All Fields]) AND (“Breathe (Sheff)” [Journal] OR “breathe” [All Fields])) OR (golden [All Fields] AND minute [All Fields]) OR ((“infant, newborn” [MeSH Terms] OR (“infant” [All Fields] AND “newborn” [All Fields]) OR “newborn infant” [All Fields] OR “neonatal” [All Fields]) AND (“resuscitation” [MeSH Terms] OR “resuscitation” [All Fields]))) AND mortality [All Fields].

### 2.2. Inclusion Criteria

The term newborn was considered as a gestational age of greater than or equal to 37 weeks of gestation and up to 30 days of life. The term preterm newborn was considered as a gestational age of fewer than 37 weeks and up to 30 days of life and/or 40 weeks of corrected age at term.Studies, whose objective was to evaluate the effect of the implementation of the HBB program in private or public health institutions (hospitals or clinics), in low and middle-income countries or scenarios.Reporting data on mortality and morbidity outcomes.Clinical trials, quasi-experimental studies, and observational studies.

### 2.3. Exclusion Criteria

Literature reviews such as systematic, integrative, and/or narrative reviews; a summary of conferences and correspondence to the editor.Poster presentations, conferences, and/or abstracts only.

### 2.4. Outcomes

The primary outcome was newborn mortality, defined as death in the period from birth to the first 28 days of life. The secondary outcomes were intrapartum mortality, defined as the birth of a viable fetus with a gestational age greater than 22 weeks or birth weight greater than 500 g, an Apgar score of 0 at minutes one and five, without signs of maceration, and presenting fetal heart sounds at the entrance and onset of labor; mortality in the first 24 h; early mortality, understood as the death of the newborn in the first 7 days of life, and late mortality, which was defined as death between 8 and 28 days of life. Morbidity outcomes were the effect on perinatal asphyxia, intraventricular hemorrhage, necrotizing enterocolitis, neonatal sepsis, bronchopulmonary dysplasia, and length of stay in the neonatal unit.

### 2.5. Screening and Inclusion of Studies

The initial search and selection of studies were carried out independently by two researchers (PA, AC). Initial results were compared, and discrepancies were resolved by consensus with a third researcher (SA). To define their final entry into the systematic review, the articles identified as relevant by screening were retrieved in full text for in-depth reading independently by the two researchers. Again, the discrepancies were resolved by consensus with a third investigator (SA).

### 2.6. Data Extraction and Synthesis

Information on the characteristics of the study was extracted in terms of bibliometric data (author, year, and country of publication) and data relevant to the study (type of health institution included and geographic area, study methods, characteristics of the included newborn cohort, methods of how the implementation of the program was carried out, educational strategy, and outcomes evaluated). This information was extracted independently by the reviewers. Differences were resolved through discussions and consensus. The assessment of the risk of bias in the observational studies was carried out with the Robins I checklist [18].

### 2.7. Statistical Analysis

Odds ratios (OR) with a 95% confidence interval (CI) were used as a measure of effect size. Random effects models were used to account for different sources of variation among studies. Heterogeneity was assessed using Q of Cochrane, which determined if the variability of the effects was greater than those expected by chance, and the I2 statistic test was used to rate the degree of heterogeneity as none <25%, low 25–49%, moderate 50–74%, and high ≥75%. If heterogeneity existed, subgroup analyses according to the quality and risk of bias of the studies were performed to determine its source. Report and publication bias was assessed by examining the degree of asymmetry in a funnel plot, and funnel plot symmetry was assessed with the Egger’s test. STATA 14 software was used for analyses.

## 3. Results

### 3.1. Characteristics of Included Studies

A total of n = 6380 studies were identified. After deleting duplicates and initial screening, n = 22 studies were selected as potentially eligible. Finally, ten articles were chosen, and when performing the snowball strategy, one additional document was found. Therefore, for qualitative synthesis, eleven studies were included, while for meta-analysis, ten studies were included (Figure 1). The main causes of exclusion were another type of intervention, different outcomes, and types of study.

Regarding the study design, n = 8 were before and after studies, two were prospective cohort studies, and one was a clinical trial [19]. The implementation of the intervention was carried out in health institutions (private or public hospitals and rural or urban hospitals) and focused on the training of health personnel (nurses, doctors, and students) as well as midwives during vaginal births and cesarean sections.

The studies included a total of n = 412,741 infants, of which n = 106,317 were preterm newborns. However, not all studies report gestational age at birth. Additionally, it was observed that the implementation of HBB was carried out under different strategies and took different training times between the different cohorts. Finally, all studies were assessed overall, and subgroup mortality was given by intrapartum in the first 24 h, early, and late mortality. Regarding the morbidity reported in the included studies, only two studies [20,21] reported it, and it was in relation to the outcome of perinatal asphyxia (Table 1).

### 3.2. Assessment of Quality and Risk of Bias of the Studies

The risk of bias in the studies was moderate to critical, especially in the domains of confusion, measurement of results, and selection bias. This was because the domain of confusion, population, and/or outcome was not well defined. Likewise, the measurement and selection of the results were not well reported (Table 2).

### 3.3. Meta-Analysis Results

The studies evaluated overall mortality and subgroups. The meta-analysis indicates that there is a reduction in the risk of overall death (OR 0.67; 95% CI 0.57, 0.8) Figure 2a, intrapartum stillbirth death (OR 0.62; 95% CI 0.51, 0.75) Figure 2b and first-day neonatal mortality (OR 0.57, 95% CI 0.41, 0.8) Figure 2c. Late mortality did not change with the intervention (Figure 2e).


Regarding the morbidity outcome, only two studies [20,21] evaluated the effect on perinatal asphyxia. The meta-analysis of these studies shows a tendency to reduce this outcome with the implementation of the HBB program (OR 0.04; 95% CI 0.00, 0.98). However, heterogeneity is very high, and the confidence interval is wide (Figure 3). Rule et al. [20] showed a high decrease in the risk of asphyxia with the implementation of HBB, but this study has a high risk of bias, so we believe that the results were overestimated and are the cause of heterogeneity. Msemo et al. [21] was more accurate and had a low risk of bias.

Of the overall mortality sensitivity analyses performed, six were at low risk of bias and four were at high risk of bias, finding that the quality of the studies does not affect the outcome (Figure 4). Finally, the funnel plot shows symmetry in most of the studies, ruling out publication bias in the studies (Figure 5).

## 4. Discussion

The systematic review and meta-analysis studied the effect of the implementation of the HBB program, in low- and middle-income countries, on neonatal mortality and morbidity. We found that the implementation of the program in the health institutions of these countries decreased neonatal mortality, especially intrapartum stillbirth, first-day neonatal mortality, and first-week neonatal mortality, with no observed effect on late neonatal mortality. On the other hand, the only morbidity outcome reported in the included studies was perinatal asphyxia, which showed a reduction in this outcome with the implementation of the HBB program.

These results are in line with those reported by other authors. For example, Morris [30], in a systematic review without meta-analysis, reports that the implementation of the HBB program seems to have benefits in reducing intrapartum neonatal mortality in the first week of life. Similarly, the meta-analysis of Versantvoor et al. [31] demonstrated that HBB impacts intrapartum stillbirth, and early neonatal mortality (first-day and first-week neonatal mortality), without effect on late mortality. Nevertheless, in the present study, we found and included a larger number of studies in the literature because we decided to include studies in low- and middle-income countries and studies that inform morbidity outcomes, while the study of Versantovoor assessed only the effects in low-income countries and mortality. Given that Colombia is classified as having middle economic income and part of the neonatal mortality occurs in this type of country, we wanted to expand the effect of the HHB program in middle-income countries.

Intrapartum and early neonatal deaths can explain 5 million neonatal deaths in the world, mainly in low-income countries. In the face of this challenge, the implementation of the HBB program at the country level could have a great effect on reducing neonatal mortality [32]. In addition, to achieve the potential of the program, an educational strategy for staff training is not enough: government efforts are required for an adequate implementation of the program [33]. In line with this, we propose, as observed in the qualitative review of these studies, that the effects on mortality reduction can be explained in part by the educational strategies and national implementation measures used at the country level that developed the studies for the implementation of the program. Therefore, The HBB program can then be proposed as a prevention strategy in newborn care and intervention that can contribute to achieving the millennium development goals, allowing a decrease in neonatal mortality in countries with a high incidence. Therefore, it is proposed to continue advancing in the integration of government and welfare actors for the actual implementation of the program in these countries and scenarios with limited economic resources.

Likewise, the effect of reducing mortality can be explained by the training of the personnel in charge of the newborn during birth, which offers the necessary skills to respond to intrapartum and birth complications, while late neonatal mortality may be associated with other causes unrelated to childbirth and/or late birth complications. Although the studies included in the meta-analysis show great heterogeneity in the trained personnel (doctors, nurses, midwives, and health profession students), it is also true that the evidence shows that the training of personnel in resuscitation is a strategy that decreases neonatal mortality [34]. Studies of the HBB program have shown that it has an impact on improving and retaining the skills and knowledge necessary for basic neonatal resuscitation [35]. It has a special effect on improving bag-mask ventilation and uses in the first minute, increasing the number of babies who receive it adequately when they need it [27,36]. This is relevant, given that about 95% of newborns manage to start breathing with adequate positive pressure bag-mask ventilation [37]. This strategy could not only impact the newborn child without vital signs (intrapartum stillbirth), but all newborns when it is applied in an appropriate way, decreasing the chance of dying in the first 24 h; this fact is important because the window of greatest mortality after a cardiorespiratory arrest or asphyxia occurs during the first 24 h.

Although heterogeneity was found in the type of staff trained in the program, it should also be recognized that the HBB strategy was implemented in health caregivers, midwives, doctors and nursing staff, including students and trainees in these areas; it focuses on all levels of health personnel, and the results obtained make the strategy attractive for obtaining necessary skills and access to all levels of training, in all the studies that meet the criteria.

On the other hand, the only morbidity outcome reported in these studies was perinatal asphyxia. It is important to note that there are no other types of pathologies recorded in the studies, such as length of stay in the neonatal intensive care unit, necrotizing enterocolitis, neonatal sepsis, bronchopulmonary dysplasia, intraventricular hemorrhage in the medium and long term in relation to neurodevelopment, so we suggest that future studies could take these outcomes into account.

Finally, although the effect on intrapartum and early mortality is important to meet the SDG, complementing the impact of combining it with other programs and/or neonatal support interventions such as when essential care for the newborn is recommended [38]. Therefore, we also propose evaluating the effect on early and late mortality in research studies with the establishment of programs in conjunction with others of interest in neonatal health, such as post-arrest stabilization and transport courses, such as STABLE^®^, Acute Care of at-Risk Newborns (ACoRN^®^) and with a properly established referral network allowing timely access to complex care to adequately continue post-arrest newborn care.

The study has some limitations. The first is the heterogeneity of the studies, which limits the validity of the results. We believe that heterogeneity is due to the lack of data in some studies, such as gestational age, educational strategy, and staff. Second, it is the low quality of the studies that limit the recommendations and extrapolation. The strengths of this study lie in the inclusion of recent literature with effects on low- and middle-income countries, where the highest neonatal mortality occurs, and in the evaluation of the impact on outcomes other than mortality that largely explain the burden of disease in neonates who survive birth complications and asphyxia; although only asphyxia is reported as an outcome, we believe that these data open new research opportunities that strengthen the HBB program. Finally, the methodology used for the systematic review and data extraction was its strength.

## 5. Conclusions

In conclusion, the HBB program is effective in reducing intrapartum stillbirth and early mortality (first day and first week). Given that the highest concentration of neonatal mortality occurs in this period and due to perinatal asphyxia, the HBB program has great potential to contribute to achieving the MDGs.

## Figures and Tables

**Figure 1 medicina-58-01567-f001:**
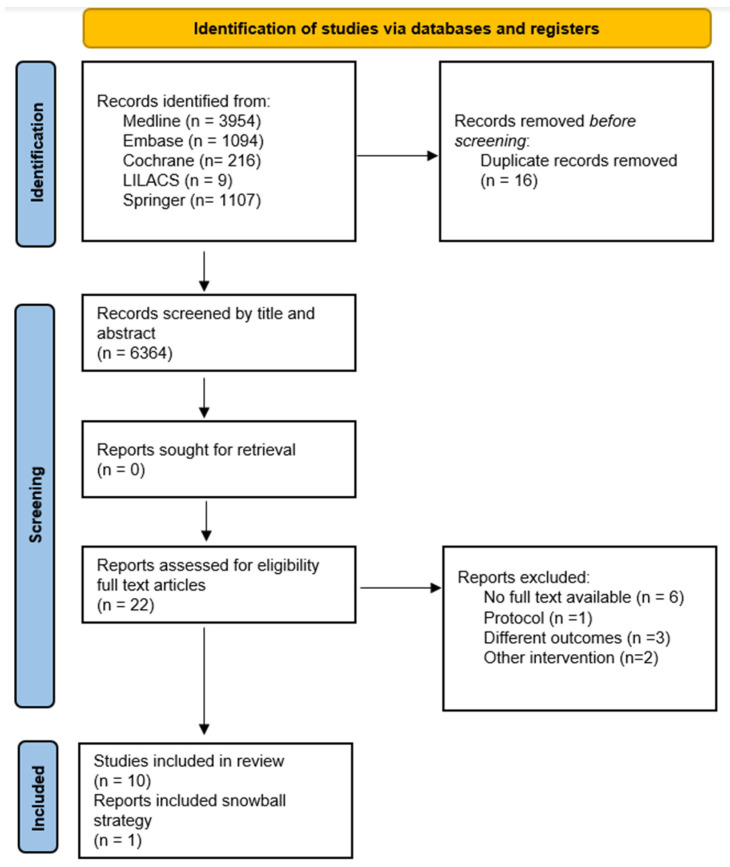
Study flowchart.

**Figure 2 medicina-58-01567-f002:**
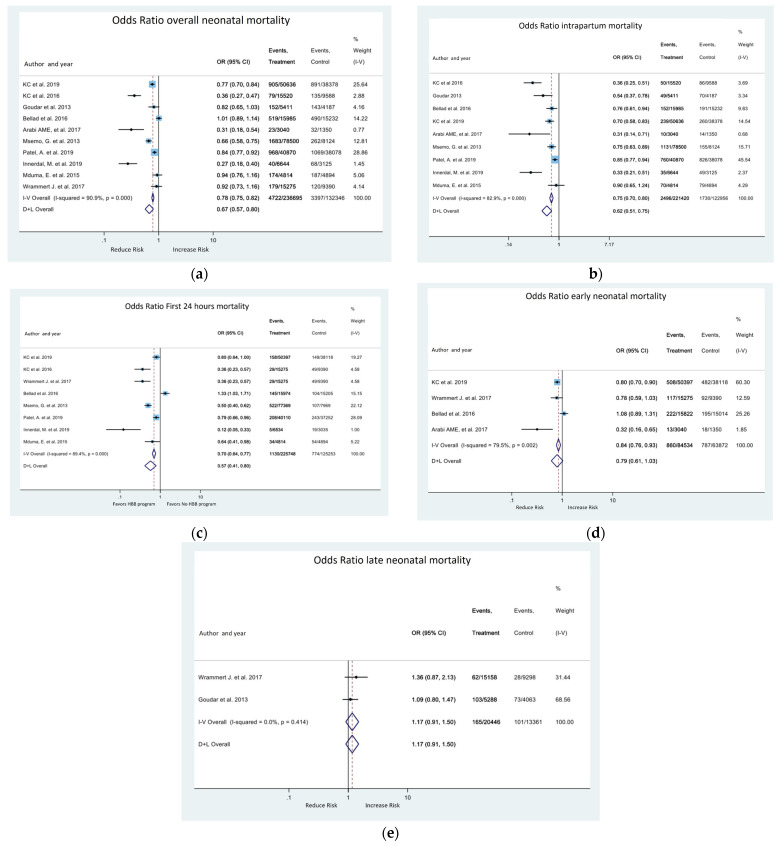
Forest plot for the effect of HBB program on neonatal mortality; (**a**) overall neonatal mortality; (**b**) intrapartum stillbirth mortality; (**c**) first-day neonatal mortality; (**d**) first week (early) neonatal mortality; (**e**) late neonatal mortality.

**Figure 3 medicina-58-01567-f003:**
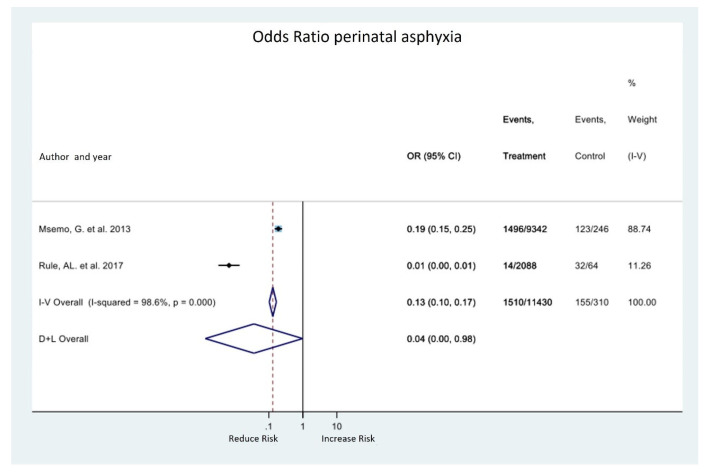
Forest plot for effect of HBB program on neonatal morbidity.

**Figure 4 medicina-58-01567-f004:**
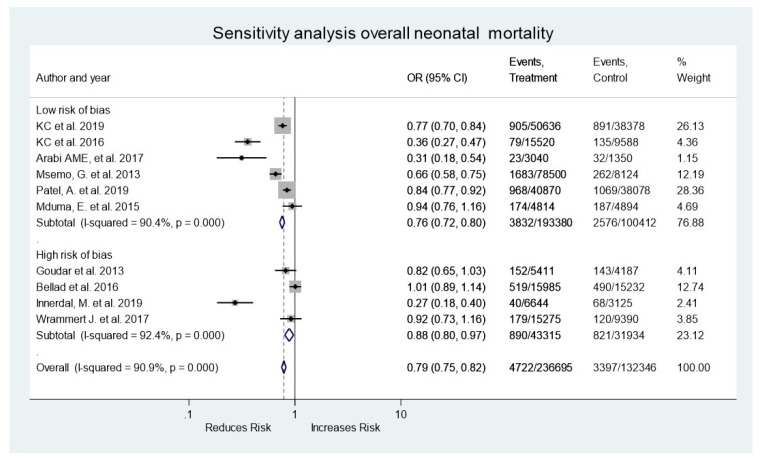
Forest plot for sensitivity analysis for overall neonatal mortality.

**Figure 5 medicina-58-01567-f005:**
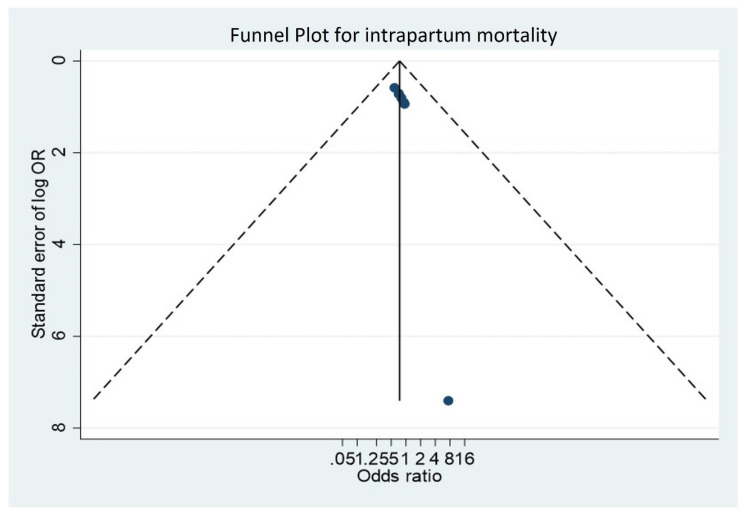
Funnel plot (asymmetry test).

**Table 1 medicina-58-01567-t001:** Characteristics of the included studies.

Author, Year, Country	Design	Duration of Study	Sample and Place	Objective	Intervention-Implementation Strategy	Measured Outcomes
Msemo 2013 Tanzania [21]	Before and after	2 years	86.621 (8124 before and 78,500 after) 8 hospitals in Tanzania	To determine whether the implementation of HBB improves the basic skills of those attending deliveries, including the application of mask bag ventilation, and whether it reduces early neonatal mortality by 50% and death rate.	For 6 to 9 months: The principal investigator and trainers conducted a one-day training of healthcare providers at each hospital	Overall mortality, intrapartum mortality, 24 h mortality and asphyxia
Mduma 2015 Tanzania [22]	Before and after	2 years (2010–2012)	9807 (4894 before and 4812 after) 1 hospital in Tanzania	To assess whether frequent and brief HBB simulation training would affect clinical practice and reduce 24 h neonatal mortality.	Training in FBOS HBB simulation. One-day trainings for everyone who works in the delivery room. Monthly training sessions of 40 min duration. The practical sessions focused on the immediate basic care of stabilization and resuscitation intervention.	Overall mortality, intrapartum mortality, 24 h mortality
Rule 2017 Kenya [20]	Before and after	21 months (1/2014–9/2015)	4117 (2106 before and 2011 after) 1 hospital in Bomet, Kenya	Describe a study that uses quality improvement. The hospital’s Neonatal Task Force identified high rates of asphyxia at birth (BA) as a quality gap. With the implementation of HBB, they sought to reduce hospital BA rates by 50% over a six-month period.	An HBB coach joined the team for one year to train its members in the HBB methodology. Prior to initial training, reference practices in the delivery room were observed, staff members were interviewed, and task force members were trained as HBB teachers.	Asphyxia
Patel 2019 India [23]	Before and after	2 years (2011–2013)	78,948 (38,078 before and 40,870 after) 13 hospitals in Nagpur, India	To assess perinatal mortality at day 1 in facility deliveries before and after HBB implementation	HBB training of instructors who then trained birth attendants, introduction of a multifaceted follow-up program, and retraining of delivery attendants after six months. They were instructed to reanimate all non-macerated births, including those considered fresh stillbirths.	Overall mortality, intrapartum mortality, 24 h mortality
Innerdal 2019 Mali [24]	Before and after	3 years (2015–2018)	9769 (3125 before and 6644 after) 1 hospital in Mali and 13 district health centers.	Reduce neonatal mortality in Mali by introducing HBB.	The implementation of the first edition of HBB was 44 sessions, of 1 or 2 days. The evaluation of the training was carried out with a written test before and after the sessions. Then they trained in the second edition of HBB with a duration of 2 to 3 days and weekly repetition training was introduced.	Overall mortality, intrapartum mortality, 24 h mortality
KC et al., 2019 Nepal [19]	Randomized controlled trial	18 months (4/2017–10/2018)	89.014 (control 38.378, intervention 50,636) 12 public hospitals in Nepal	Phased implementation of a quality improvement package for neonatal resuscitation (HBB) in hospitals in Nepal	Implementation of a quality improvement package in neonatal resuscitation that includes facilitation strategies, training, weekly meetings, and information dissemination visits.	Intrapartum mortality, 24 h mortality, early mortality
KC et al., 2016 Nepal [25]	Prospective cohort study	14 months (7/2012–9/2013)	25,108 (control 9588, intervention 15,520) 1 tertiary hospital in Nepal	Improve adherence to the Helping Babies Breathe neonatal resuscitation protocol by using a quality improvement cycle	HBB protocol training, weekly review meetings, daily skills checks, use of self-assessment checklists, and refresher training.	Overall mortality, intrapartum mortality, 24 h mortality
Bellad et al., 2016 India y Kenia [26]	Before and after	24 months (1/2011–10/2013)	70,704 (before 35,595 and then 35,109) Belgaum: 33 centers Nagpur: 15 centers Kenya: 23 centers	To assess the impact of implementing a package of HBB interventions and monitoring in select health facilities representing a large proportion of births and perinatal mortality rate at sites in India and Kenya	Master trainer training and training of childbirth care teams. It included assessment of HBB knowledge and skills before and after training courses and updates 6 months later.	Overall mortality, intrapartum mortality, 24 h mortality, early mortality.
Wrammert J. et al., 2017 Nepal [27]	prospective cohort study	15 months (7/2012–9/2013)	24,665 (control 9390 and intervention 15,275) 1 tertiary hospital in Kathmandu	Describe the timing and causes of neonatal deaths in hospital before and after HBB training at a maternity health center in Nepal	Evaluation of the effect of HBB training on neonatal mortality rates	General mortality, 24 h mortality, early mortality, late mortality.
Goudar et al., 2013 India [28]	Before and after	11 months (10/2009–09/2010)	9598 (before 4187 and then 5411) District hospitals in Karnataka, India, and urban hospitals in Belgaum	To assess the efficacy of HBB training in reducing stillbirths and neonatal mortality rate	Model of training and teaching and skills and practice, coaches were trained, including discussion, practice, and simulation. Training to trainers was continued and learning assessments were applied.	Overall mortality, intrapartum mortality, late mortality.
Arabi AME, et al., 2017 Sudan [29]	Before and after	24 months	4390 (before 1350 and after 4390) 6 rural medical centers in east Nile	Community-based intervention (village midwives) to assess the impact of HBB on neonatal mortality	Trainers at HBB instructed midwives, included simulator training kit and teaching materials, then weekly post-HBB follow-up	Intrapartum mortality, early mortality

**Table 2 medicina-58-01567-t002:** Summary of Risk of Bias in Included Studies.

Article/Domain	Confusion	Participants Selection	Classification of Interventions	Deviations and Interventions	Lack of Data	Measurement of Results	Result Selection Reported	Global	Risk
Ashish KC 2016 [25]									Moderate
Bellad et al., 2016 [26]									Serious
Wrammert et al., 2017 [27]									Critical
Goudar et al., 2013 [28]									Serious
Ashish KC 2019 [19]									Moderate
Arabi AME, et al., 2017 [29]									Moderate
Msemo G, et al., 2013 [21]									Moderate
Patel A, et al., 2019 [23]									Moderate
Rule AL, et al., 2017 [20]									Serious
Innerdal M, et al., 2019 [24]									Serious
Mduma E, et al., 2015 [22]									Moderate

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
