# Peer review of "Effect of the Helping Babies Breathe Program on Newborn Outcomes: Systematic Review and Meta-Analysis"

_medicina, 2022, doi:10.3390/medicina58111567_

Round 1

Reviewer 1 Report

I congratulate the authors on writing this manuscript evaluating the impact of implementing the Helping Babies Breathe Program in Low-middle income countries. 

Author Response

Thanks to the reviewer for the comments in favor of improving the paper. A single file with suggested grammatical and style corrections is attached.

Reviewer 2 Report

Content is relevant to the title. Aim of the study is clearly stated. The design of the study is appropriate. Appropriate statistical techniques have been selected. Tables and Figures are appropriately entitled and appropriate representations of the data. All references are included in the text. Conclusions are valid. 

This manuscript could be accept for publishing.

Author Response

Thanks to the reviewer for the comments in favor of improving the paper. A single file with suggested grammatical and style corrections is attached

Reviewer 3 Report

The results published in this work are predictable, but useful nonetheless. There is no doubt that adequate training of medical staff in newborn resuscitation will have a major impact on early neonatal mortality. The quality of early resuscitation will affect late neonatal mortality indirectly. These results apply mainly, but not only, to the developing countries of the world. In these conditions, a number of socio-economic factors play their role, which is mainly signaled by the disproportionately high percentage of prematurity (over 25%) as the main factor of perinatal pathology. Emphasis of work on systematic training of personnel in postnatal resuscitation It is still a current topic.

Author Response

Thanks to the reviewer for the comments in favor of improving the paper.